# A 52-to-57 GHz CMOS Phase-Tunable Quadrature VCO Based on a Body Bias Control Technique

**Seongmin Lee, Yongho Lee**  **and Hyunchol Shin \*** 

Department of Electronic Convergence Engineering, Kwangwoon University, Seoul 01897, Republic of Korea; tnehf333@kw.ac.kr (S.L.); dldyd91@kw.ac.kr (Y.L.)
* Correspondence: hshin@kw.ac.kr; Tel.: +82-2-940-5553

**Abstract:** This paper presents a 52-to-57 GHz CMOS quadrature voltage-controlled oscillator (QVCO) with a novel I/Q phase tuning technique based on a body bias control method. The QVCO employs an in-phase injection-coupling (IPIC) network comprising four diode-connected FETs for the quadrature phase generation. The I/Q phase error is calibrated by controlling the body bias voltage offset of the QVCO's four core FETs. This technique effectively covers a wide range of I/Q phase error between −13.4° and +10.7°. It also minimally induces the unwanted variations in the phase noise, current dissipation, and oscillation frequency, which were found to be only 0.4 dB, 0.07%, and 36 MHz, respectively. After the IPIC-QVCO, a phase-tunable two-stage LO buffer employing a 3-bit switched-capacitor bank was added for additional phase tuning, leading to the extension of the phase tuning range up to −22.7–+20.0°. The proposed QVCO is implemented in a 40 nm RF CMOS process. The measured results show that the QVCO covers a frequency band from 52.4 to 57.6 GHz while consuming 26.2 mW. The phase noise and the figure-of-merit of the QVCO are −91.8 dBc/Hz at 1 MHz offset and −172.4 dBc/Hz, respectively. We also realized a fully integrated 55 GHz quadrature RF transmitter employing the phase-tunable QVCO and LO generator. The effectiveness of the proposed phase-tunable LO generator was confirmed by verifying the image rejection ratio (IRR) calibration at the RF output.

**Keywords:** quadrature voltage-controlled oscillator (QVCO); in-phase injection-coupled (IPIC); I/Q phase mismatch calibration; millimeter wave; 60 GHz; CMOS



## 1. Introduction

With the fast-growing demand for 60 GHz band wireless applications such as radar sensors [1,2] and short-range connectivity [3], this band has been attracting lots of research and development efforts. To produce single-chip RF transceivers for 60 GHz wireless applications, a quadrature transceiver architecture is widely adopted. This is because 60 GHz wireless communication usually employs quadrature modulation and demodulation schemes such as quadrature amplitude modulation (QAM) to achieve a high data rate [4–6]. One of the challenges in designing a quadrature transceiver is the I/Q phase imbalance in the I/Q LO signals. The I/Q phase error is known to cause severe performance degradations in the signal-to-noise ratio (SNR) and image rejection ratio (IRR). Therefore, an efficient and precise calibration technique for the I/Q phase error is essentially needed for the quadrature LO generation in CMOS technology.

The quadrature LO generation in CMOS can be realized in various manners. A $2 \times f_o$ voltage-controlled oscillator (VCO) and a subsequent divide-by-two circuit are widely employed due to the simple design and high I/Q accuracy. Unfortunately, however, this approach would not be practical for the 60 GHz band, because the required 120 GHz VCO would be too challenging. An RC-CR polyphase filter connected after a fundamental $f_o$ VCO is another possible approach, as can be found for 28 GHz transceivers [7,8]. However, in the 60 GHz band, improving the I/Q mismatch and insertion loss of the

RC-CR filter imposes too great a challenge. Meanwhile, in order to alleviate the VCO design requirements in the quadrature LO generator, an injection-locked frequency tripler (ILFT) has been reported [9–12]. In [9,10], a tunable phase shifter was employed to generate quadrature LO signals at $f_o/3$, which were then tripled by a subsequent ILFT. This approach, however, would make it difficult to achieve accurate I/Q phase tuning, since the ILFT triples not only the frequency but also the phase error. Another approach is to employ a phase-tunable quadrature ILFT, as can be found in the 60 GHz [11] and 28 GHz [12] bands. These employ a phase-tunable bottom-series-coupled QVCO (BS-QVCO), which tunes the I/Q phase by controlling the QVCO's varactors at the fundamental frequency $f_o$ in a fine step. However, this approach suffers from a narrow locking range due to the inherent injection-locking characteristics.

In contrast to the above methods, direct use of a fundamental QVCO with phase-tuning capability should show the best advantages of simple circuitry and a wide tuning range. However, the phase error can be as high as several degrees unless it is accompanied by proper phase-tuning circuitry [13–20]. The parallel-coupled fundamental QVCOs (P-QVCOs) in 60 GHz [13] and 50 GHz [14] did not report any phase-tuning technique; thus, a phase error as poor as 2.5 degrees was found in [14]. Magnetic-coupled QVCOs [15–17] also reported about 1.5 degrees of phase error at 60 GHz without any phase-tuning circuitry. On the other hand, a superharmonic-coupled 60 GHz QVCO (SH-QVCO) in [18] employed an independent varactor tuning at the two VCO cores to compensate for the I/Q phase error. However, the phase error was as poor as $\pm5$ degrees. In [19], a 45 GHz SH-QVCO employed a tunable transconductance ($g_m$) stage in the coupling network to compensate for the I/Q phase error within 0.3 degrees, but its phase-tuning range was too limited at only +0.1–+2.5 degrees.

Meanwhile, it has been found that an in-phase injection-coupled (IPIC) QVCO with a diode-based coupling network [20,21] is an effective and efficient fundamental QVCO architecture due to its class-C-like operation and low phase-noise performances. Indeed, the 60 GHz IPIC-QVCO in [20] showed excellent phase matching, but no phase-tuning circuitry was reported. Hence, in this work, we present a 60 GHz IPIC-QVCO with a novel I/Q phase-tuning capability based on a body bias control of the VCO core FETs. A theoretical analysis is described to prove how the body bias control can induce the phase tuning in the IPIC-QVCO. Implemented in a 40 nm CMOS process, the 60 GHz IPIC-QVCO showed a wide I/Q phase tuning range of $-13.4$–$+10.7$ degrees. Furthermore, with an additional phase-tunable LO buffer attached after the IPIC-QVCO, the phase-tuning range was extended to $-22.7$–$+20.0$ degrees. In order to prove the I/Q phase-tuning performance, the complete QVCO and LO generator was fully integrated in a 60 GHz quadrature RF transmitter [6], demonstrating that the IRR at the RF transmitter output is dramatically improved by the proposed phase-tuning technique.

## 2. Circuit Design

### 2.1. Phase-Tunable In-Phase Injection-Coupled (IPIC) QVCO

Figure 1 shows the circuit schematic of the phase-tunable IPIC-QVCO. It comprises two cross-coupled negative-$g_m$ VCO cores and a coupling network in between to create the desired quadrature phase. The coupling network comprises four diode-connected n-FETs ($M_{c1-4}$). It injects coupling currents into the VCO cores at the aligned phase, realizing the in-phase injection coupling of the two VCO cores and leading to the desired quadrature VCO operation. The frequency tuning characteristics are achieved through the 3-bit discrete-tuned switched-capacitor bank $C_t<2:0>$ for producing eight sub-band tuning curves and the MOS varactor diodes $C_v$ for producing a continuously tuned curve within each sub-band tuning curve. The switched-capacitor bank circuit has a conventional design, as can be found in many previous papers [12,22,23].

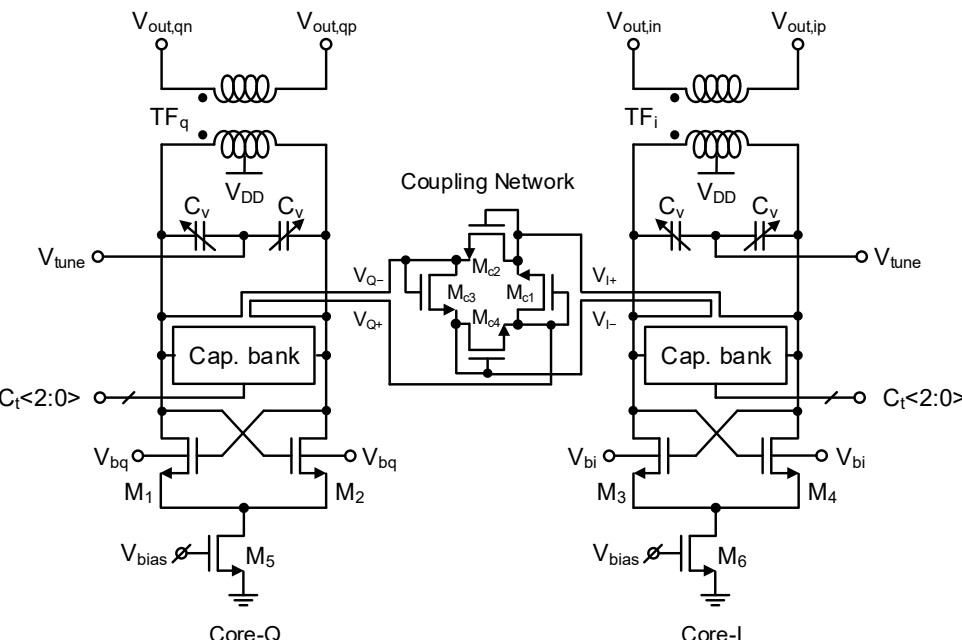

**Figure 1.** Proposed QVCO schematic.

The I/Q phase error should be caused by the mismatches of the VCO cores and the coupling network. Without any calibration, the I/Q phase error typically reaches a few degrees in millimeter-wave CMOS QVCOs [11,14–20]. Hence, this work proposes a novel I/Q phase-tuning technique in order to suppress the phase error to a negligible sub-degree level. The proposed tuning method intentionally offsets the individual oscillation frequencies of the two VCO cores by tuning the body bias voltages $V_{bi}$ and $V_{bq}$ of the n-FETs $M_{3,4}$ and $M_{1,2}$, respectively. From the initial value of 0.25 V, the $V_{bi}$ and $V_{bq}$ are changed by the same amount but in opposite directions, enabling the desired I/Q phase calibration without a significant change in the oscillation frequency.

Before we further describe the details of the I/Q phase-calibration technique, let us first examine the operating principle of the IPIC-QVCO. Figure 2a illustrates one of the diode-connected n-FETs, $M_{c2}$, of the coupling network shown in Figure 1. As shown in Figure 1, the gate terminal of $M_{c2}$ is connected to the $V_{I+}$ node of VCO core I, while the source terminal is connected to the $V_{Q-}$ node of the other VCO core Q. As shown in Figure 2b, $V_{I+}$ has the same amplitude and frequency as $V_{Q-}$, but its phase is shifted by 90 degrees. Therefore, $V_{gs}$ of $M_{c2}$ (=$V_{I+} - V_{Q-}$) is formed at 45 degrees behind $V_{I+}$. Since $M_{c2}$ turns on only when $V_{gs}$ exceeds the threshold voltage, $M_{c2}$'s drain current $I_{c2}$ should be created in phase with $V_{gs}$. Similar to $I_{c2}$, $M_{c1}$'s drain current $I_{c1}$ is 45 degrees behind $V_{Q+}$. Thus, $I_{c1}$ should be in phase with $M_{c1}$'s $V_{gs}$, and $I_{c1}$ is 90 degrees behind $I_{c2}$.

Figure 2c visualizes the current flow vector of the coupling network. The injection current $I_{inj,i+}$ flowing from the coupling network into the core VCO is given by $I_{c1} - I_{c2}$. Since $I_{c1}$ has the same amplitude with $I_{c2}$ and is 90 degrees behind that of $I_{c2}$, the phase of $I_{inj,i+}$ should be 45 degrees behind $I_{c1}$. Consequently, $I_{inj,i+}$ is in phase with the node voltage $V_{I+}$ and the drain current $I_{d4}$ of the core FET $M_4$, as well as with the tank current $I_{tank4}$. Through the same analysis, we can know that the four signals $V_{I+}$, $V_{Q+}$, $V_{I-}$, and $V_{Q-}$ are set to differential I/Q signals.

The above operating principle of the IPIC-QVCO was verified by circuit simulations. Figure 2d–f show the simulated time-domain voltage and current waveforms at the IPIC-QVCO. Figure 2d shows $I_{c1}$, $I_{c2}$, and $I_{inj,i+}$. It is confirmed that $I_{c1}$ and $I_{c2}$ have a phase difference of 90 degrees, while $I_{inj,i+}$ is 45.9 degrees behind $I_{c1}$. In Figure 2e, it can be observed that $I_{tank4}$ and $I_{d4}$ are aligned well, while $I_{inj,i+}$ shows slight misalignment with $I_{tank4}$ and $I_{d4}$. In spite of the slight misalignment, the quadrature phase relationships between $I_{c1}$ and $I_{c2}$ (as shown in Figure 2d) and between $V_{I+,-}$ and $V_{Q+,-}$ (as shown in

Figure 2f) prove that the in-phase injection locking of the QVCO works properly to generate the quadrature output signals at the QVCO. The whole operation can be confirmed by observing the four output voltages $V_{I+}$, $V_{Q+}$, $V_{I-}$, and $V_{Q-}$ in Figure 2f (with an amplitude of 0.62 V and a frequency of 58 GHz). We can confirm that the IPIC-QVCO produces the desired differential quadrature phase output signals properly.

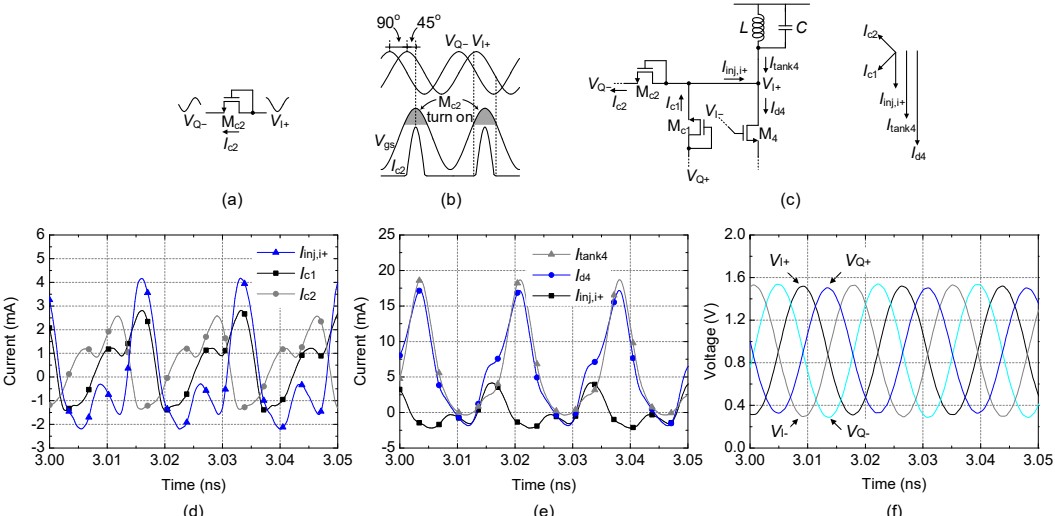

**Figure 2.** Operating principle of the in-phase injection coupling in the QVCO: (**a**) diode-connected coupling FET and (**b**) its turn-on operation. (**c**) Current flow diagram at the VCO core and the coupling network. Simulated current and voltage waveforms: (**d**) $I_{c1}$, $I_{c2}$, and $I_{inj,i+}$; (**e**) $I_{inj,i+}$, $I_{tank4}$, and $I_{d4}$; and (**f**) final quadrature output voltages $V_{I+}$, $V_{Q+}$, $V_{I-}$, and $V_{Q-}$.

When we apply a body bias voltage at the QVCO, the body leakage of the FETs must be carefully monitored. Figure 3 shows the simulated leakage current flowing through the body terminal of the core's FET when the QVCO oscillates with the body bias voltage swept from 0 to $V_{DD}$. Both the peak and average values of the body leakage current are plotted for the sake of comparison. As can be seen, when the body bias voltage exceeds 0.5 V, the body current begins to rapidly increase. This will result in an unacceptable increase in the dynamic current consumption of the QVCO, which, in turn, will lead to undesirable performance degradations in the phase noise and I/Q phase error of the QVCO. Thus, we set the maximum limit of the body bias voltage to 0.5 V. Through simulations, we found that the QVCO's current consumption is increased by 20 μA when the body bias voltage is set to 0.5 V, which is only a 0.07% change compared to the nominal consumption of 29.1 mA.

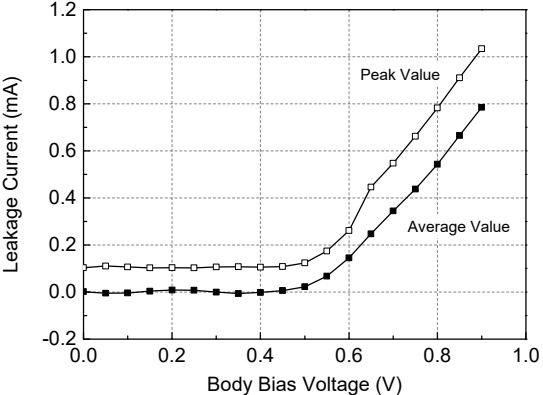

**Figure 3.** Simulated body leakage current.

Yi et al. [20] derived an analytic relation of the I/Q phase error $\Delta\varphi$ in the QVCO with respect to the oscillation frequency offset $\Delta\omega_o$ of the two VCO cores. $\Delta\varphi$ is given by

$$\Delta\varphi = K_o \frac{\Delta\omega_o}{\omega_o} \tag{1}$$

where $K_o$ is a damping coefficient and $\omega_o$ is the nominal oscillation frequency of the QVCO. The damping coefficient $K_o$ is given by

$$K_o = \frac{\sqrt{2}QI_o(1-m)}{g_{mC}V_{tC}} \tag{2}$$

where $Q$ is the LC tank quality factor, $m$ is the coupling strength, $I_o$ is the tail current of the QVCO, and $g_{mC}$ and $V_{tC}$ are the transconductance and threshold voltage of the coupling diode-connected FET, respectively. The coupling strength $m$ is expressed as

$$m = \frac{I_{inj}}{I_{osc}} \tag{3}$$

where $I_{inj}$ is the injection current and $I_{osc}$ is the VCO's tank current. Since the tank current is supplied by the tail current source $I_o$ through the hard-limiting switching operation of the core FETs, and the tank $Q$ is sufficiently high, only the fundamental component of the tank current can be taken into account, which is $2/\pi \times I_o$.

Based on the Equations (1)–(3), we can calculate the theoretical phase-tuning range of the QVCO. By conducting DC and transient simulations of the designed VCO circuit shown in Figure 1, we first estimated the required parameter values of $\omega_o$ (58.5 GHz), tank $Q$ (7.6), $I_o$ (13.8 mA), and $I_{inj}$ (3.0 mA). Since $I_{osc}$ is $2/\pi \times I_o$, $I_{osc}$ is 8.78 mA. Then, the coupling strength $m$ was computed to 0.342. In addition, the $g_{mC}$ and $V_{tC}$ of the coupling diodes $M_{c1-4}$ were found through transient simulations to be 9.32 mS and 454.5 mV at 58.5 GHz, respectively. Finally, the damping coefficient $K_o$ was computed to 23.04 using Equation (1). As a next step, we found through transient simulations that the VCO core's oscillation frequency increases by 400 MHz when the body bias voltage increases by 0.5 V. Finally, we computed the theoretical phase-tuning range to be 9.03 degrees using Equation (1). It should be noted that $K_o$ and $\omega_o$ in Equation (1) remain almost unchanged against the body bias voltage control. Therefore, we can conclude that the body bias control only changes $\Delta\omega_o$ (and, therefore, the phase error $\Delta\varphi$) according to Equation (1).

The theoretical prediction given by Equations (1)–(3) was verified through circuit simulations. Figure 4a shows the simulated frequency offset $\Delta\omega_o$ and phase error $\Delta\varphi$ against the body bias voltage offset $\Delta V_b = V_{bi} - V_{bq}$. In this design, the initial values of $V_{bi}$ and $V_{bq}$ were set at +0.25 V, so that the initial $\Delta V_b$ and $\Delta\omega_o$ were all set to zero. Then, the $V_{bi}$ and $V_{bq}$ were tuned by the same amount but in opposite directions, while the maximum $V_{bi}$ and $V_{bq}$ were limited to 0.5 V. As a result, $\Delta V_b$ was tuned between $-0.5$ and $+0.5$ V. As shown in Figure 4a, the frequency offset $\Delta\omega_o$ and the phase error $\Delta\varphi$ were found to change from $-390$ to $+390$ MHz and from $-7.1$ to $+8.7$ degrees, respectively, across the entire range of $\Delta V_b$. Figure 4b compares the simulated and theoretically calculated phase error $\Delta\varphi$ versus the frequency offset $\Delta\omega_o$. The theoretical data were calculated from Equation (1), and the simulated data were obtained from Figure 4a. We found that the calculated data agreed very well with the simulated results, confirming that the proposed I/Q phase-tuning calibration technique is valid and effective. Meanwhile, we investigated how the phase noise is affected by the body bias control. Figure 4c shows the simulated phase noise against $\Delta V_b$. It demonstrates that the phase noise changes only by 0.3 dB across the $\Delta V_b$ tuning range. Thus, it confirms that our proposed phase-tuning technique minimally impacts the phase noise performance of the QVCO.

The robustness of the proposed phase-tuning technique against the PVT corner variations was examined. Figure 5 shows the simulated I/Q phase error, oscillation frequency, and phase noise versus the body bias offset $\Delta V_b$ under seven PVT corners. The seven

corners include two process corner conditions of SS and FF, two voltage corner conditions of 0.81 V and 0.99 V, and two temperature corner conditions of −40 °C and +80 °C, from the nominal condition having the typical process corner of TT, typical $V_{DD}$ of 0.9 V, and typical temperature of +27 °C.

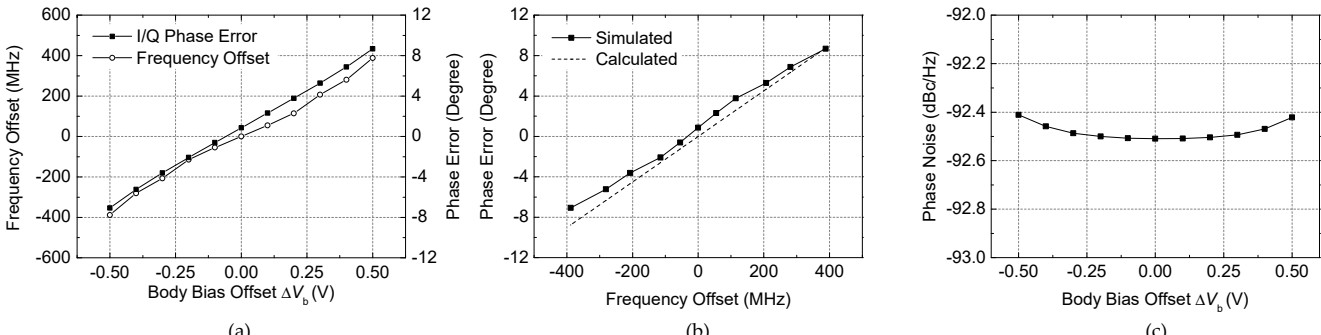

**Figure 4.** (**a**) Simulated frequency offset and I/Q phase error against the body bias voltage offset $\Delta V_b = V_{bi} - V_{bq}$ ($V_{bi} = V_{bq} = 0.25$ V when $\Delta V_b = 0$ V). (**b**) Simulated and theoretically computed I/Q phase error against the frequency offset. (**c**) Simulated phase noise against $\Delta V_b$.

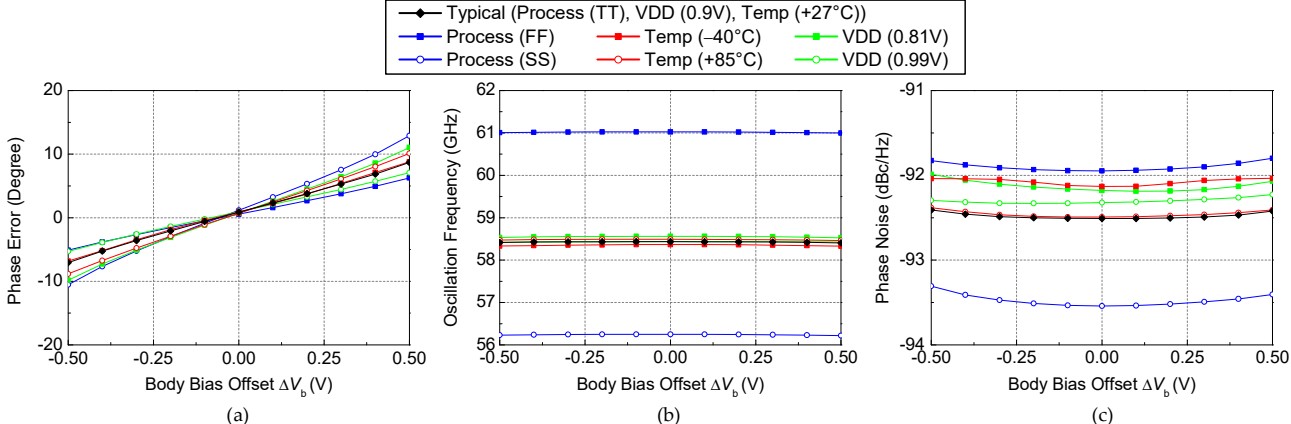

**Figure 5.** Results of seven PVT corner simulations: (**a**) I/Q phase error, (**b**) oscillation frequency, and (**c**) phase noise, all against the body bias offset $\Delta V_b$.

Figure 5a shows that the phase-tuning range varies with respect to seven corners, with typical, maximum, and minimum ranges of −7.1–+8.7, −10.6–+12.9, and −5.1–+6.3 degrees, respectively. We found that even the minimum tuning range is acceptable for our 60 GHz LO generator design application when we add an additional phase-tunable LO buffer, as described in Section 2.2. Figure 5b shows that the oscillation frequency would change by as much as 2.5 GHz across the FF and SS corners; however, it does not change significantly against $\Delta V_b$ (which is only 36 MHz). This indicates that the proposed body bias control technique does not induce significant oscillation frequency and performance variations. Figure 5c also shows that the body bias control does not cause significant changes in the phase noise performance (only 0.2–0.3 dB), although it may change more against the corners.

### 2.2. Quadrature LO Generator with the Proposed QVCO and Phase-Tunable Buffer

A complete quadrature LO generator including the proposed QVCO was designed by adding a phase-tunable LO buffer after the QVCO. The architecture of the quadrature LO generator is shown in Figure 6. The VCO buffer isolates the QVCO's LC tank from the subsequent loading effect. The two-stage phase-tunable LO buffer gives additional phase tunability, thereby extending the overall I/Q phase-tuning performance significantly.

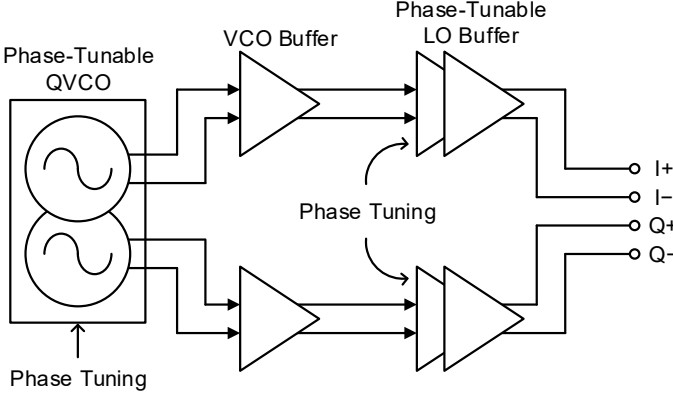

**Figure 6.** Block diagram of the entire quadrature LO generator.

Figure 7 shows the schematic of the VCO buffer and the phase-tunable LO buffer. They were designed in a fully differential stage with a tail current source in order to improve the common-mode noise immunity. Note that the transformer $TR_1$ shown in Figure 7 indicates the coupling transformer $TR_{i,q}$ at the VCO output in Figure 1. The small signal gain of the buffer stage was designed as high as 21.9 dB to guarantee sufficient swing at the output. Note that the phase-tuning capability of the LO buffer was realized by using 3-bit discrete tuned capacitor banks in the two stages. The same approach for the phase-tunable LO buffer design can be found in [8,24].

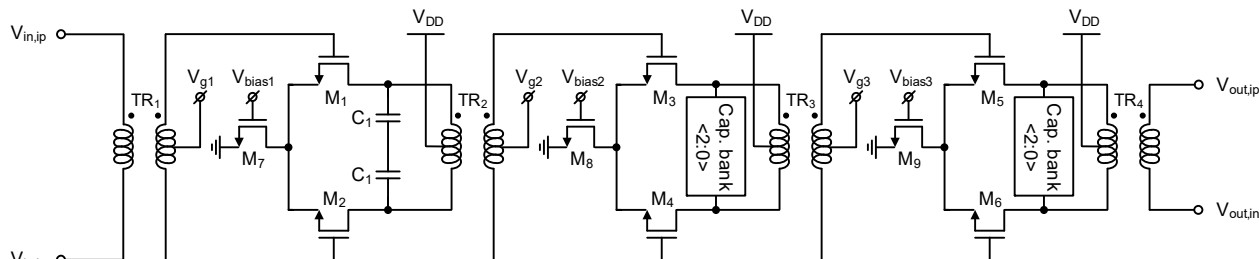

**Figure 7.** Schematic of the single-stage VCO buffer and the phase-tunable two-stage LO buffer.

The current dissipation is 5.6 mA for the first-stage VCO buffer, and 10.1 mA for the second- and third-stage LO buffer.

Figure 8 shows the simulated performance of the overall phase-tuning characteristics of the entire quadrature LO generator. The I/Q phase tuning is first achieved by controlling the body bias offset $\Delta V_b$ between $-0.5$ and $+0.5$ V, as noted by the (0,0)-curve in Figure 8. This shows the phase-tuning range of $-13.4$ to $+10.7$ degrees. It is interesting to note that it becomes slightly wider than the original tuning range of the QVCO alone without the buffer, as shown in Figure 4b. Now, in addition to the $\Delta V_b$ tuning, the phase-tunable LO buffer is simultaneously tuned using the 3-bit switched-capacitor banks. Since we have two stages for the phase-tunable buffer, the tuning code can be set to 64 different codes between (0,0) and (7,7). In Figure 8, only nine selected states (the lowest code 0, the middle code 4, and the highest code 7 for the first and second stages) are plotted. This clearly shows that the phase-tuning curve is shifted up and down with respect to the capacitor bank code, while the total tuning range against $\Delta V_b$ does not change much. The total tuning range from codes 0 to 7 for each LO buffer was found to be about 4–5 degrees, resulting in the overall tuning range of the QVCO including the LO buffer of 42.7 degrees (from $-22.7$ to $+20.0$ degrees). Thus, we can see that the proposed LO generator block gives a sufficiently wide tuning range. Note that the tuning accuracy is determined by the $\Delta V_b$ tuning resolution. We can estimate that the tuning accuracy will be $0.37°$ by assuming a 6-bit voltage digital-to-analog converter (VDAC) is used for $\Delta V_b$ tuning, as reported in

our other works [24,25]. In this work, however, the VDAC circuit was not included in the QVCO, whereas the body bias voltage was directly tuned by using external voltage sources.

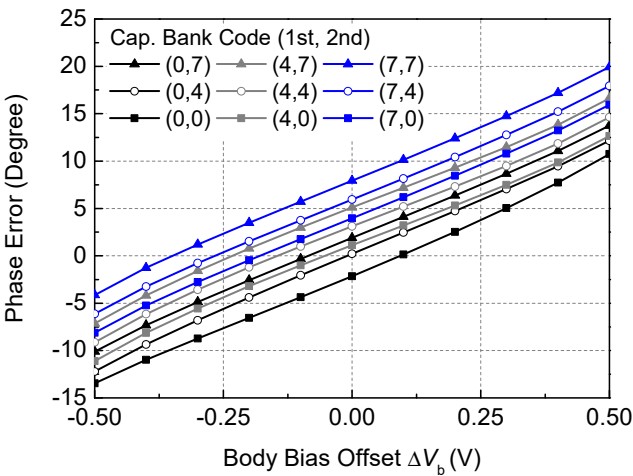

**Figure 8.** Simulated overall phase-tuning characteristics of the quadrature LO generator.

## 3. Implementation Results

The IPIC-QVCO integrated circuit was fabricated in a 40 nm RF CMOS process. A die micrograph of the QVCO is shown in Figure 9a. Note that additional circuitry of a two-stage VCO buffer and a divide-by-eight frequency divider was included for test purposes. The total die area, including the pads, is $540 \times 1150~\mu m^2$, while the QVCO core occupies $260 \times 137~\mu m^2$, and the first- and second-stage VCO buffers occupy $240 \times 132$ and $110 \times 120~\mu m^2$, respectively. The divide-by-eight frequency divider consists of three cascaded divide-by-two circuits, with individual areas of $268 \times 130$, $390 \times 190$, and $460 \times 170~\mu m^2$, respectively. The QVCO consumes a current of 29.1 mA from a supply voltage of 0.9 V. The VCO buffers consume 28.7 mA, and the three divide-by-two circuits consume 11.9 mA, 7.2 mA, and 5.8 mA, respectively. The measurement setup for the QVCO is shown in Figure 9b. The QVCO output spectrum was measured by using a Keysight N9030B spectrum analyzer.

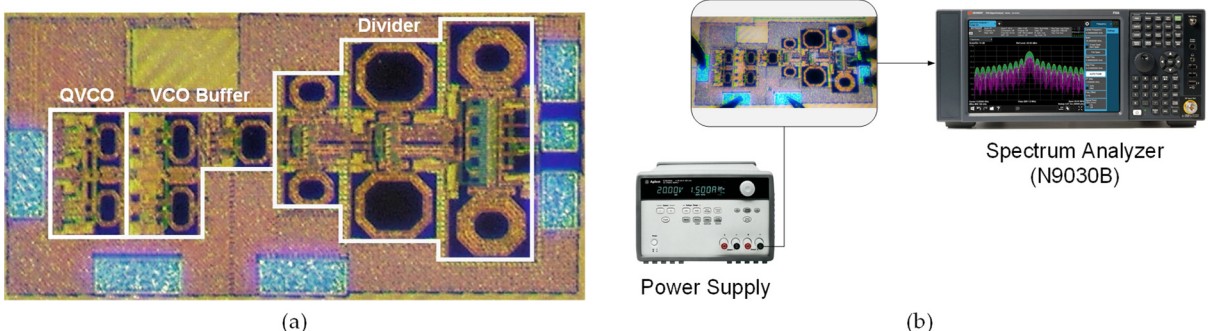

(a)

(b)

**Figure 9.** (**a**) Die micrograph of the QVCO followed by a two-stage VCO buffer and a divide-by-eight frequency divider added for test purposes. (**b**) Measurement setup.

Figure 10 shows the output spectrum of the QVCO measured using a Keysight N9030B spectrum analyzer. Note that the measured frequency of 6.8 GHz should be multiplied by 8 to obtain the QVCO oscillation frequency of 54.4 GHz.

Figure 11a shows the measured QVCO frequency tuning range. With the 3-bit sub-band tuning, the eight discrete tuning curves are clearly exhibited. The tuning range of the QVCO is from 52.4 to 57.6 GHz, giving a total tuning range of 9.2 GHz. A slight frequency downshift from the simulation results is observed, which should be attributed to unexpected parasitic LC components in the fabricated circuit.

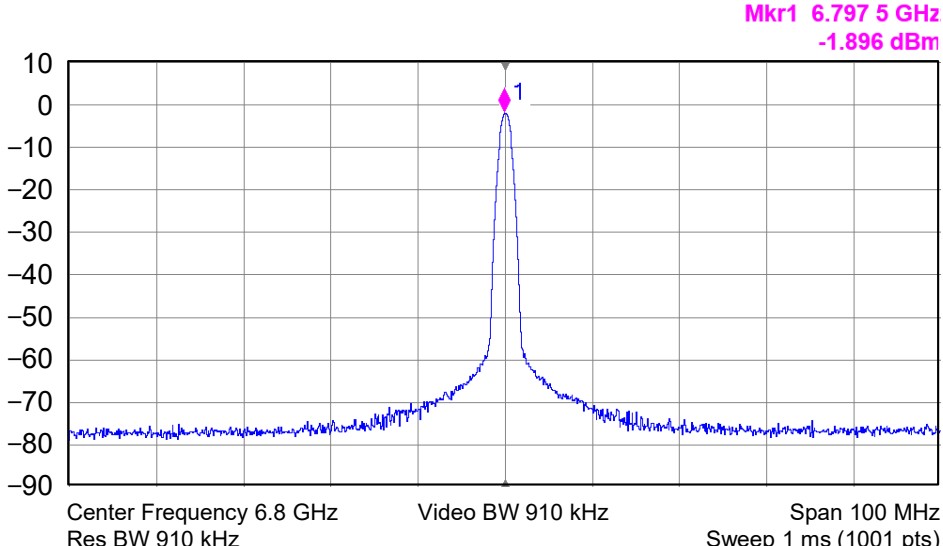

**Figure 10.** Measured output spectrum of the QVCO after the divide-by-eight.

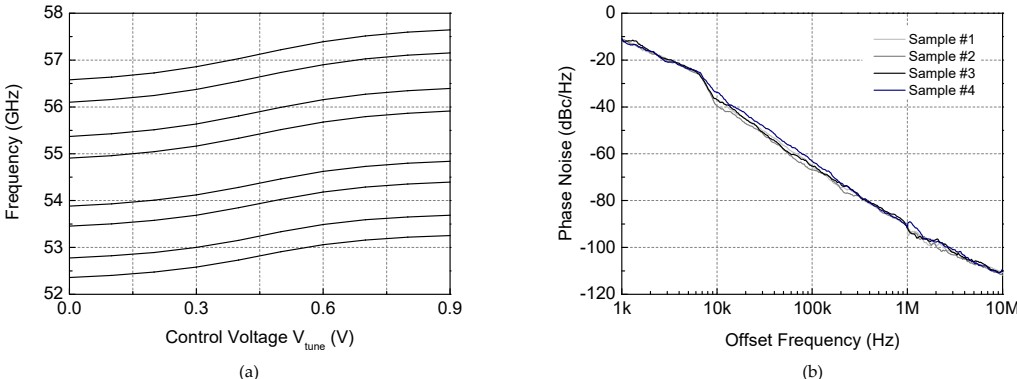

(a)

(b)

**Figure 11.** Measured results: (**a**) frequency tuning range; (**b**) phase noise of four samples.

Figure 11b shows the phase noise performances of the QVCO measured from four samples. The measured phase noises were observed between −90.2 and −91.8 dBc/Hz at a 1 MHz offset for a 54.6 GHz oscillation frequency. This corresponds to the VCO's figure-of-merit (FoM) of −170.8−−172.4 dBc/Hz. At 100 kHz and 10 MHz offsets, the phase noises from the four samples were also found to be −63.2−−66.9 dBc/Hz and −109.3−−112.0 dBc/Hz, respectively. The corner frequency dividing the $1/f^3$ and $1/f^2$ region was found to be less than 2 MHz, indicating a low flicker noise upconversion process.

For more complete verification of the phase-tuning operation of the proposed circuit, the quadrature LO generator comprising the IPIC-QVCO, VCO buffer, and phase-tunable LO buffer was fully integrated into the author's 55 GHz quadrature RF transmitter [6]. Figure 12a,b show a die micrograph and the measurement setup of the fabricated RF transmitter, respectively. The total die area is $600 \times 1380 \ \mu m^2$.

The fabricated transmitter chip was mounted on a printed circuit board. DC and low-frequency signals were fed through wire-bonded pads, and RF signals were fed and probed by V-band on-wafer probes. For the output spectrum measurement, the QVCO was first set to oscillate at 56 GHz, and the baseband I/Q signals at 300 MHz were applied to the baseband input ports. Then, we could recognize that the transmitter's output creates three tones: the LO leakage at 56 GHz, the desired tone at 56.3 GHz, and the unwanted image tone at 55.7 GHz.

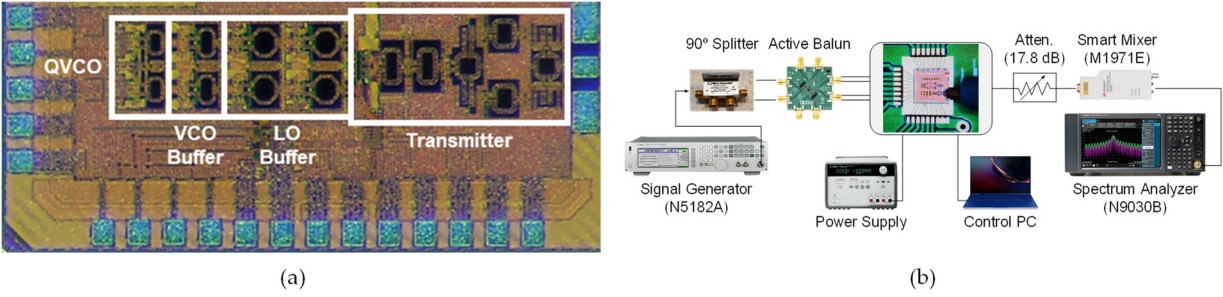

|     |     |
| :-: | :-: |
| (a) | (b) |

**Figure 12.** (**a**) Die micrograph of the fully integrated quadrature RF transmitter employing the proposed QVCO and phase-tunable LO buffer. (**b**) Measurement setup.

Figure 13 demonstrates that the IRR at the RF transmitter output is improved by the phase-tuning capability of the QVCO and LO generator. Figure 13a shows the output spectrum before the phase calibration. The desired and image tones appear exactly 300 MHz away from the LO leakage tone, and the IRR is only −13.2 dB, which is unacceptably high. Figure 13b shows the worst case when the I/Q phase calibration is set to the worst condition, in which the IRR becomes as poor as −9.4 dB. Finally, Figure 13c shows the best condition obtained by setting the optimal I/Q phase calibration. The overall calibration procedure was as follows: we first performed the coarse tuning through the LO buffer capacitor bank calibration and, subsequently, further performed the fine tuning through the QVCO body bias calibration. We observed that the IRR was improved by up to −20.4 dB. It was also noted that the magnitude and frequency of the desired signal remained almost unchanged after the calibration, which implies that the proposed phase-calibration technique effectively provides a wide-ranging and highly accurate phase calibration without introducing significant changes in the LO amplitude and frequency. Thus, the proposed phase-calibration technique can ensure high-quality signal transmission at the 60 GHz millimeter-wave band.

Table 1 summarizes the performances found in this work and compares them with other previous works. The previous mmWave QVCOs such as the P-QVCO [14], magnetic-coupled QVCOs [15–17], and IPIC-QVCO [20] did not report any I/Q phase-calibration technique in their QVCOs. Thus, no tuning performances were found in their publications. Meanwhile, two SH-QVCOs reported that a certain form of I/Q phase-calibration technique was realized in their QVCOs [18,19]. The 60 GHz SH-QVCO in [18] added additional varactors in each VCO core's LC tank and tuned the varactor capacitances to calibrate the I/Q phase. However, they reported a slightly poor phase-calibration accuracy of ±5 degrees. The 45 GHz SH-QVCO in [19] employed a tunable $g_m$-stage-based coupling network for the phase calibration. However, it reported a very limited tuning range of less than 2.5 degrees.

In contrast, this work was based on a novel combined approach of the body bias voltage control for the QVCO and the switched-capacitor tuning for the LO buffer. It demonstrated a very wide tuning range of −22.7–+20.0 degrees with sub-degree accuracy. In order to measure the phase tunability in this work, we have shown that the entire I/Q LO generator comprising the QVCO and LO buffer is integrated with a 60 GHz quadrature RF transmitter, and the image rejection ratio at the RF transmitter output was examined to verify the phase-tuning performances. It is interesting to note that [18] incorporated a downconversion mixer and verified the phase-tunability at the downconverted baseband signal, while [19] incorporated an I/Q self-mixer to generate a DC voltage proportional to the I/Q phase error. Although the verification methods are all different, they are acceptable to verify the phase-tunability in the millimeter-wave circuit design. Meanwhile, the power consumption and phase noise of this work were found to be slightly worse than in [18,19], although it should be noted that this is not attributed to the proposed calibration technique.

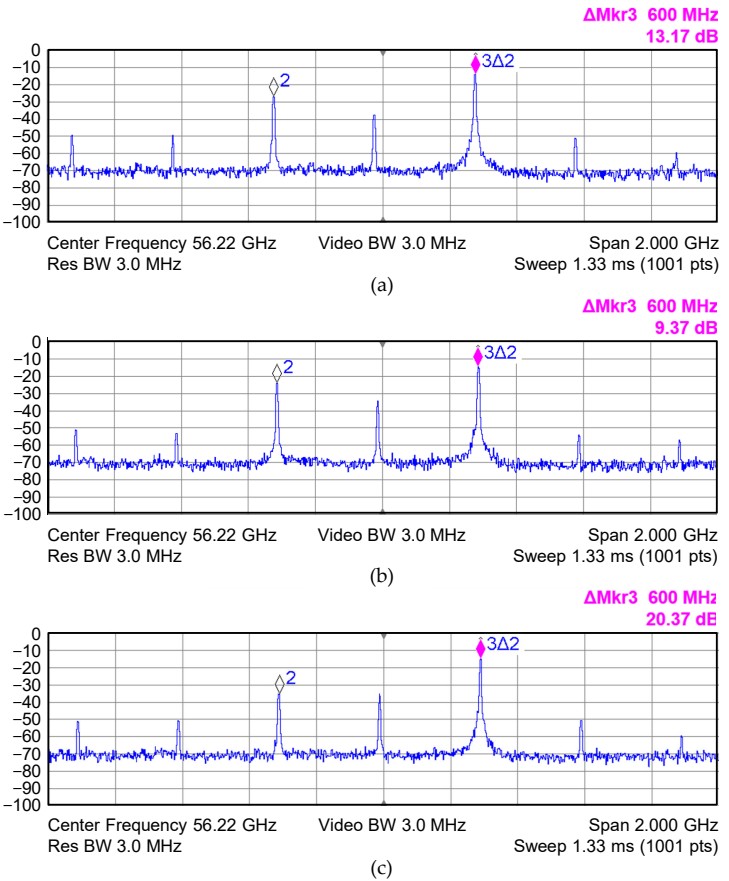

**Figure 13.** Measured 56 GHz output spectrum at the RF transmitter output with the phase calibration set to (**a**) the default condition, (**b**) the worst condition, and (**c**) the best condition.

**Table 1.** Performance summary and comparison with the recent mmWave QVCOs.

|  | **This Work** | **[19]** | **[18]** | **[20]** | **[17]** | **[16]** | **[15]** | **[14]** |
|---|---|---|---|---|---|---|---|---|
| Frequency Range (GHz) | 52.4–57.6 | 37.5–45.1 | 53.8–63.3 | 57.9–68.3 | 51.7–56.6 | 48.8–62.3 | 56.0–60.4 | 55.6–59.1 |
| Frequency-Tuning Range (%) | 9.5 | 18.4 | 16.2 | 16.6 | 9.1 | 24 | 7.5 | 4.1 |
| I/Q Phase-Tuning Method | Body bias | Coupling $g_m$ stage | Tank varactor | None | None | None | None | None |
| I/Q Phase-Tuning Range (Degrees) | −13.4–+10.7 (−22.7–+20.0) [3] | +0.1–+2.5 | −25.0–+25.0 | – | – | – | – | – |
| Phase Noise @1 MHz (dBc/Hz) | −91.8 | −94.3 | −94.5 | −94.2 | −95.5 | −89.8 | −96 | −83.2 |
| FoM [1] (dBc/Hz) | −172.4 | −177.4 | −178.4 | −179.6 | −176.4 | −172.7 | −177.9 | −165.8 |
| FoM$_T$ [2] (dBc/Hz) | −172.0 | −182.7 | −182.6 | −184.0 | −175.5 | −180.4 | −175.5 | −161.5 |
| Supply Voltage (V) | 0.9 | 0.75 | 0.9 | 1.2 | 0.8 | 1.2 | 1.0 | 1.2 |
| Power Dissipation (mW) | 26.2 | 8.4 | 14 | 11.4 | 24 | 16 | 22 | 18 |
| CMOS Process (nm) | 40 | 28 | 40 | 65 | 65 | 65 | 65 | 90 |

[1] FoM = phase noise − $20\log(f_{out}/f_{offset})$ + $10\log(P_{diss}/1\,\text{mW})$, [2] FoM$_T$ = FoM − $20\log$ (tuning range (%)/10%), [3] QVCO with phase-tunable LO buffers.

## 4. Conclusions

A 55 GHz CMOS IPIC-QVCO with a noble I/Q phase-calibration technique was designed and implemented in a 40 nm RF CMOS process. For a wide and precise I/Q phase-tuning performance, the IPIC-QVCO employs body bias tuning for the QVCO core FETs. Theoretical and simulated analyses confirmed the operating principle and tuning performances of the proposed method. The overall tuning range was extended by additional phase-tunable LO buffers. The combined phase calibration covers a wide

tuning range of 42.7 degrees, with sufficiently good accuracy of less than 1 degree. The prototype QVCO integrated circuit was fabricated in 40 nm RF CMOS process and verified through chip measurements. The fully integrated 55 GHz quadrature RF transmitter IC with the proposed LO generator successfully demonstrated significant improvements of IRR performances at the RF output. This proves that the proposed IPIC-QVCO and phase-tunable LO buffer should be instrumental for millimeter-wave CMOS quadrature LO generator circuit design.

**Author Contributions:** Conceptualization, Y.L. and H.S.; methodology, Y.L. and H.S.; validation, S.L., Y.L. and H.S.; formal analysis, S.L. and H.S.; investigation, S.L. and H.S.; data curation, S.L. and Y.L.; writing—original draft preparation, S.L., Y.L. and H.S.; writing—review and editing, S.L. and H.S.; visualization, S.L. and Y.L.; supervision H.S.; project administration, H.S.; funding acquisition, H.S. All authors participated in all other aspects. All authors have read and agreed to the published version of the manuscript.

**Funding:** This research was supported by the National Research Foundation (NRF) of Korea under Grant 2020R1A2C1008484.

**Data Availability Statement:** Data sharing not applicable.

**Acknowledgments:** This work was conducted during a sabbatical year from Kwangwoon University in 2021. The computer-aided circuit design tools of this work were supported by the IC Design and Education Center of Korea (IDEC).

**Conflicts of Interest:** The authors declare no conflict of interest. The funders had no role in the design of the study; in the collection, analyses, or interpretation of data; in the writing of the manuscript; or in the decision to publish the results.

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
