# Peer review of "A 52-to-57 GHz CMOS Phase-Tunable Quadrature VCO Based on a Body Bias Control Technique"

_electronics, doi:10.3390/electronics12122679_

Round 1

Reviewer 1 Report

The letter reported the application and combinatorial optimization of IPIC-QVCO and phase-tunable LO buffer. The results showed it was instrumental for millimeter-wave CMOS quadrature LO generator circuit design. An innovative viewpoint and method, However, there are still the following issues that need to be emphasized. Here's a minor Revision for you.

1What influences the increase of body current in Fig.3?

2Why is there no error bar in Figure 4, and is the accuracy high when repeated multiple times? Such problems also arise elsewhere.

3How is the chip area determined based on the aforementioned principles and structure? Do you have any specific considerations?

4 What are the operating steps for the I/Q phase calibration

5Is the testing environment for other methods the same as yours, and is it comparable for phase calibration? What are the specific materials, processes, and production processes

6What instruments are used for system on a chip testing?

the Quality of English Language is fine.

Reviewer 2 Report

The paper is well written. Theory is supported by measurement results.  I have only one minor comment: 

As body biasing is very sensitive to temperature variation, The effect of temperature on circuit performance must be investigated. 

Reviewer 3 Report

This paper is very appealing; the idea is novel and is experimentally shown to be successful.  I have some comments:

Please include a photo of the test setup and, if possible. the photo of the bonding of the chip.

The images in Fig.2, and especially the captions, are very small in size, which makes the explanation related to this figure difficult to follow. Please improve it.

In Fig. 5. the legend looks small and blurred.

I think the fact that the phase noise does not change substantially with body voltage is something important to note, and should appear before the last paragraph on page 6. Perhaps a graph like those in Fig. 4(a) would be worthwhile (it could be included in Fig. 4, as Fig.4(c)).

The curve in Fig. 10 is blurred, as is the grid, but the numbers on the axes are not. If this is an image capture of the HS, do not edit the axes. Otherwise, improve the quality of the curve by processing the data with, for example, Matlab. The same idea happens in Fig. 13.

Regarding the bank of capacitors: What is the effect of the switches on the operation of the circuit?

Regarding the power shown in  Table 1: as it is 2 or 3 times higher than other references, I think this has to be discussed.

In Table 1, the phase noise of the circuit is somewhat higher than in other implementations. This is due to the correction, or without the correction, this would be even worse.

Round 2

Reviewer 2 Report

No comment